# High *Wnt2* Expression Confers Poor Prognosis in Colorectal Cancer, and Represents a Novel Therapeutic Target in BRAF-Mutated Colorectal Cancer

**DOI:** 10.3390/medicina59061133

**Published:** 2023-06-12

**Authors:** Huan Liu, Lihua Zhang, Ye Wang, Rendi Wu, Chenjie Shen, Guifang Li, Shiqi Shi, Yong Mao, Dong Hua

**Affiliations:** 1Wuxi Medical College, Jiangnan University, Wuxi 214000, China; 6202809010@stu.jiangnan.edu.cn (H.L.); 6202809026@stu.jiangnan.edu.cn (R.W.); 6212809029@stu.jiangnan.edu.cn (C.S.); 6212809017@stu.jiangnan.edu.cn (G.L.); 6212809063@stu.jiangnan.edu.cn (S.S.); 2Department of Oncology, The Affiliated Hospital of Jiangnan University, Wuxi 214000, China; 3Department of Gastrointestinal Surgery, Zhongshan Hospital of Xiamen University, School of Medicine, Xiamen University, Xiamen 361004, China; lihuazhang@xmu.edu.com; 4Department of Urology, Shanghai Changzheng Hospital, Naval Medical University, Shanghai 200003, China; wy15026489059@smmu.edu.cn; 5Wuxi People’s Hospital, Wuxi 214000, China

**Keywords:** colorectal cancer, *Wnt2*, *BRAF* mutations, prognosis

## Abstract

*Background and Objectives:* We aimed to investigate the role of *Wnt2* expression in colorectal cancer (CRC) prognosis and evaluate its potential as a therapeutic target in *BRAF*-mutated CRC. *Materials and Methods*: Exactly 136 samples of formalin-fixed paraffin-embedded CRC tissue specimens were obtained from patients who underwent surgical resection for CRC. The gene mutation status of the samples was detected using fluorescence PCR. *Wnt2* expression was detected using immunohistochemistry. Survival curves with high *Wnt2* expression and *BRAF* mutations were compared using the Kaplan–Meier method. A nomogram was constructed to determine the estimated overall survival probability. We also predicted the 3-year and 5-year survival rates for patients with high *Wnt2* expression and *BRAF* mutations. In total, 50 samples of *BRAF*-mutated CRC were collected and detected *Wnt2* expression by immunohistochemistry. The Chi-squared test was used to analyze the association between *Wnt2* expression and *BRAF*-mutated CRC. *Results*: High *Wnt2* expression and *BRAF* mutations are associated with poor prognosis of CRC. Multivariate survival analyses indicated that high *Wnt2* expression and *BRAF* mutations are significant independent predictors of CRC prognosis. Furthermore, high *Wnt2* expression was significantly associated with *BRAF*-mutated CRC, and *Wnt2* may be a potential therapeutic target for *BRAF*-mutated CRC. *Conclusions*: High *Wnt2* expression confers poor prognosis in colorectal cancer and represents a novel therapeutic target in *BRAF*-mutated CRC.

## 1. Introduction

Colorectal cancer (CRC) is one of the most common malignant gastrointestinal tumors. A White Paper on Epidemiology, Prevention, and Screening of Colorectal Cancer in China shows that CRC ranks third in the incidence of malignant tumors. According to the 2019 China Health Statistical Yearbook, CRC ranks fifth among the top 10 malignant tumor mortality rates in China, with a mortality rate of 7.25 (1/100,000), including 8.19 (1/100,000) for males and 6.26 (1/100,000) for females. By 2030, there will be more than 2.2 million newly diagnosed cases and 1.1 million deaths, and the global CRC burden is estimated to increase by approximately 60% [1]. As the early detection of CRC is still challenging, most patients are diagnosed at an advanced stage [2]. Although significant progress has been made in treating advanced CRC in recent years, the 3-year and 5-year survival rates remain very low [3]. Therefore, potentially effective molecular biomarkers and treatment indicators for CRC diagnosis and prognosis are needed. These can help with the early detection, monitoring, and treatment monitoring of patients with CRC, improve their prognosis and survival, and promote the development of personalized treatment.

*Wnt2*, a member of the *Wnt* family located on human chromosome 7q31, encodes a secretory protein that regulates the necessary developmental process and plays a critical role in tumorigenesis [4]. Increased *Wnt2* expression has been found in human fetal lungs and the placenta, but rarely in the normal gastrointestinal tract [5]. An increasing number of studies has shown that the expression of *Wnt2* is abnormally high in various cancers, including fibroadenoma [6,7], breast cancer [8,9,10], and gastric cancer [11]. Compared with normal stroma [12], *Wnt2* is significantly overexpressed in CRC and promotes cancer cell invasion and metastasis by activating the *Wnt* signaling pathway. It is essential in maintaining the activated CAF (cancer-associated fibroblast) phenotype, which promotes angiogenesis [13]. It has also been found that CAFs-secreted *Wnt2* suppresses the DC-mediated antitumour T-cell response via the SOCS3/p-JAK2/p-STAT3 signalling cascades, thereby suppressing antitumor immunity [14].

Tumor cells accumulate genetic and epigenetic changes during growth [15]. They are composed of thousands of non-synonymous mutations that are positively associated with the occurrence and development of cancer [16]. Only a subset of these mutations can often be used as driving mutations, whereas the rest are considered random passenger mutations. These mutations accumulate and change the course of the tumors [17]. As the primary driving mutations, *BRAF* mutations are involved in the occurrence and development of various cancers. Nearly 60% of melanomas show *BRAF* mutations [18], and mutations in this gene are also found in non-Hodgkin’s lymphoma [19], CRC [20,21], thyroid papillary carcinoma [22,23], non-small-cell lung cancer [24,25,26], glioblastoma [27], and inflammatory diseases [28,29]. *BRAF* mutations, are widely believed to be associated with poor prognosis in CRC [30]. Elaine Tan et al. found that patients with *BRAF* mutations had worse OS (Overall Survival) compared with the wild type with a median survival of 18.9 months versus 33.2 months [31].

This study aimed to investigate the clinical implications of *Wnt2* expression and *BRAF* mutations in the prognosis of CRC. Moreover, we investigated the relationship between *Wnt2* and several key genes in CRC.

## 2. Materials and Methods

### 2.1. Clinical CRC Specimen and Clinical Information

This study was approved by the Ethics Committee of the Affiliated Hospital of Jiangnan University. In total, 136 patients diagnosed with CRC who underwent surgical resection at the Affiliated Hospital of Jiangnan University between June 2014 and November 2016 were selected retrospectively. Overall, 50 samples of *BRAF*-mutated CRC were retrospectively screened from January 2010 to December 2019 at the Affiliated Hospital of Jiangnan University. All patients underwent mutation testing. The exclusion criteria were as follows: (1) receiving neoadjuvant antitumor therapy such as radiotherapy, chemotherapy, or immunotherapy before surgery; (2) the type of multiple tumors or metastases is unclear; (3) the patients’ personal information is incomplete or lost to follow-up; (4) a history of other malignant tumors.

The clinicopathological features of all patients were analyzed and recorded in detail by two clinicians, and details regarding prognosis were obtained by telephone follow-up.

### 2.2. Immunohistochemical Analysis

The CRC tissue samples were fixed in 4% formalin, embedded in paraffin, and sectioned. The tissue sections were subjected to immunohistochemical analysis, deparaffinized in xylene, and hydrated with graded ethanol. Then, they were heated in sodium citrate buffer for 30 min at 100 °C for antigen repair and incubated with 3% hydrogen peroxide to block endogenous peroxidase activity. After blocking for 30 min, the cells were incubated with *Wnt2* antibody (1:100, Abcam, ab150608, Hong Kong, China) overnight at 4 °C. Subsequently, the sections were washed with phosphate buffer salt solution and incubated with an amplifying agent, polymerase (reagent A, GTVisionTM Kit, Shanghai, China), and 3, 3′-diaminobenzidine chrome developer (DAB, reagent B, C; GTVisionTM kit, Shanghai, China) for 1–3 min. Then, they were stained with hematoxylin for 60s and examined under a microscope after dehydration and sealing.

The intensity and degree of *Wnt2* immunostaining were semi-quantitatively analyzed, and the percentage of tumor cells with positive staining and intensity of staining were scored. According to the German semi-quantitative scoring system standards, each section was individually read by two associate chief physicians from the Department of Pathology, and the results were averaged.

Each pathological tissue specimen was scored according to two parameters: staining intensity and staining range, and the final score obtained by each pathologist was the product of the two parameters. The staining intensity was divided into four grades: negative = 0, weak staining = 1, moderate staining = 2, and strong staining = 3. The dyeing range was divided into: ≤5% = 0, 6–25% = 1, 26–50% = 2, 51–75% = 3, 76–100% = 4. The samples were stratified into high (scores 9–12) and low (scores 0–9) *Wnt2* expression groups.

### 2.3. Mutation Detection

The samples were tested for mutations using five mutant gene detection kits (AmoyDx, Xiamen, China) and the human *PIK3CA* gene Mutation Detection Kit (AmoyDx, Xiamen, China), according to the manufacturer’s instructions. The samples were 136 FFPE specimens from postoperative CRC patients. DNA extraction of the samples was performed using a paraffin tissue DNA extraction kit (TianGen, Beijing, China), the procedure is as follows:30 mg of paraffin tissue was collected, deparaffinized by shaking with xylene, and centrifuged at 12,000× *g* for 2 min at room temperature. The supernatant was discarded, and absolute ethanol was added and mixed by shaking. The mixture was centrifuged at 12,000× *g* for 2 min at room temperature, and the supernatant was discarded. It was allowed to stand for 5 min to fully volatilize the ethanol. An additional 200 µL GA buffer and 20 µL Proteinase K were added, mixed, and incubated at 56 °C for 1 h until the sample was completely lysed. After further incubation at 90 °C for 1 h, 220 µL GB buffer was added and mixed, then 250 µL absolute ethanol was added and mixed. The liquid was put into the adsorption column at 8000 rmp and centrifuged at room temperature for 2 min. After the waste liquid was discarded, 500 µL GD buffer was added to the adsorption column CR2 and centrifuged at 8000 rpm for 60 s. Discard the waste solution and repeat twice. The adsorption column was opened for 5 min, and then 50 µL of TE eluate prewarmed at 65 °C was added. Finally, DNA was collected.

Tissue samples were tested for mutations by Q-PCR with the five mutant gene detection kits, the procedure is as follows: the DNA of the extracted tissue samples was mixed with LMG mixed enzyme in the kit, and then the machine was detected. The reaction program is as follows: first stage, 42 °C for 5 min, 95 °C for 5 min, a cycle; second stage, 95 °C 25 s, 64 °C 20 s, 72 °C 20 s, 10 cycles; third stage: 93 °C 25 s, 60 °C 35 s, 72 °C 20 s, 36 cycles. The detection sites are as follows (Table 1):

### 2.4. Statistical Analysis

Statistical analysis was performed using the R version 3.5.3 software (version 3.5.3, http://www.r-project.org, accessed on 11 March 2019). Either the Chi-squared or Fisher’s exact test was used to assess the association between *Wnt2* expression and clinicopathological features. Moreover, Kaplan–Meier survival analysis was used to draw the overall survival (OS) curve, and the log-rank test was used to compare the differences. The Cox proportional hazards regression model was used for univariate and multivariate analyses. The Chi-squared test was used to analyze the association between *Wnt2* expression and *BRAF*-mutated CRC. *p* < 0.05 was considered statistically significant.

## 3. Results

### 3.1. Clinicopathological Features of the Patients with CRC

This study included 63 men and 73 women. Among the enrolled patients, 34 were under 60 years old, and 102 were over 60. There were 35 patients in the T1-2 stages, 101 in the T3-4 stages, 87 in the N0 stage, 49 in the N1 and N2 stages, 113 in the M0 stage, and 23 in the M1 stage. *ERBB2* mutations were found in 132 patients, *KRAS* mutations in 83, *BRAF* mutations in 14, and *PIK3CA* mutations in 112. The clinicopathological features of the patients with CRC are shown in Table 2.

### 3.2. High Wnt2 Expression and BRAF Mutations Are Associated with Poor Prognosis in Patients with CRC

To determine the expression of *Wnt2* in CRC, we evaluated its expression levels in the tissues of the 136 patients with CRC using immunohistochemical analysis. Immunohistochemistry showed that *Wnt2*-positive staining was mainly confined to the cytoplasm of the CRC cells (Figure 1A,B). Depending on the staining intensity, samples were divided into groups with high *Wnt2* expression and low *Wnt2* expression. The number of patients with high *Wnt2* expression was 37 (27.21%), and the number of those with low *Wnt2* expression was 99 (72.79%).

In addition, we evaluated the association between *Wnt2* expression and clinicopathological features, including age, sex, T stage, N stage, TNM stage, and mutation status, in different groups of patients with CRC. The difference in *Wnt2* protein expression (high vs. low) was significantly correlated with *BRAF* mutation status in patients with CRC (*p* = 0.0001), as shown in Table 2.

Next, we investigated whether high *Wnt2* expression and *BRAF* mutations are associated with CRC prognosis. The Kaplan–Meier analysis showed that CRCs with high *Wnt2* expression had significantly worse OS than those with low *Wnt2* expression (*p* = 0.00033, Figure 2A). In addition, CRCs with *BRAF* mutations also had significantly lower OS than those with the wild type (*p* < 0.0001, Figure 2B).

Univariate analysis showed that *Wnt2* (*p* < 0.0001), T stage (*p* = 0.008), M stage (*p* < 0.0001), N stage (*p* < 0.0001), TNM stage (*p* < 0.0001), and *BRAF* mutation status (*p* < 0.0001) were significantly associated with poor prognosis in CRC (Table 3). Subsequently, all important variables such as *Wnt2*, T stage, M stage, N stage, TNM stage, and *BRAF* mutation status were entered into the multivariate Cox proportional hazards model, and the results showed that *Wnt2* expression (*p* = 0.035) and *BRAF* mutations (*p* < 0.0001) were prognostic factors for poor OS in CRC (Table 3).

A nomogram was constructed to predict the 3- and 5-year survival rates. A score was matched on the score table based on the value of each factor, and the score of each factor was added to obtain the total predicted score to predict OS at 3 and 5 years (Figure 3A). *BRAF* mutations and M stage were the most important prognostic factors, followed by TNM stage, T stage, *Wnt2* expression, and *ERBB2* mutations. The nomogram validation included calibration of the primary and validation cohorts. Moreover, the Kaplan–Meier plot survival probabilities were used for the calibration. There was an agreement between the nomogram prediction and the actual OS at 3 and 5 years (Figure 3B).

### 3.3. Wnt2 Expression Was Associated with BRAF-Mutated CRC

From the correlation analysis in Table 2, *Wnt2* expression was positively correlated with *BRAF* mutation status. Considering the insufficient number of *BRAF*-mutated CRC, we collected 50 paraffin specimens from patients with *BRAF* mutations at the Affiliated Hospital of Jiangnan University from 2010 to 2019. *Wnt2* expression was analyzed using immunohistochemistry (Figure 4A,B). In 50 samples of *BRAF*-mutated CRC (contains the 14 samples of BRAF-mutated CRC from the previous 136 samples), *Wnt2* expression was high in 41 (82%) samples. Compared with 118 samples with wild-type *BRAF* mutations in the 136 samples, the number of high *Wnt2* expression was 26 (22%). Table 4 presents the correlation between *BRAF* mutations and *Wnt2* expression in CRC (*p* < 0.001). In conclusion, there was a positive correlation between the *Wnt2* expression and *BRAF*-mutated CRC, and *Wnt2* may be a potential therapeutic target for *BRAF*-mutated CRC.

## 4. Discussion

It is well established that multiple mutations of oncogenes are involved in the process of tissue transformation from normal epithelial cells to carcinomas in CRC [32]. *BRAF*, which regulates the Ras–Raf–MEK–ERK pathway and is associated with poor prognosis in CRC, was selected as a key driver, even though its mutation rate was slightly lower than that of other genes. *BRAF* mutations are also significantly associated with distant metastasis in Asian populations [33]. The prognosis of patients with *BRAF*-mutated CRC remains poor due to recurrence and metastasis [34]. Moreover, discovering the *BRAF* status is critical for understanding patient survival and prognosis. Therefore, there is an urgent need to define the molecular mechanisms of carcinogenesis and identify novel therapeutic targets for *BRAF*-mutated CRC. We analyzed the relationship between *Wnt2* expression and *KRAS-2*, *BRAF-15*, *PIK3CA-9*, *PIK3CA-20*, and *ERBB2-20*. Statistical analysis revealed that high *Wnt2* expression (*p* = 0.035) and *BRAF* mutations (*p* < 0.0001) were closely related to poor survival in patients with CRC. Furthermore, high Wnt2 expression was significantly correlated with *BRAF* mutations in CRC. This finding provides a new possible therapeutic target for the treatment of *BRAF*-mutated CRC.

*Wnt2* can increase the metastasis and invasion of fibroblasts and promote angiogenesis in CRC [35]. Meanwhile, upstream regulatory molecules of the *Wnt* signaling pathway are involved in multidrug resistance. In our study, *Wnt2* was found to be positively correlated with *BRAF* mutations; it is important to explore whether high *Wnt2* expression promotes invasion and metastasis and regulates drug resistance in patients with *BRAF* mutations.

In a recent study, anti-*Wnt2* mAb was found to significantly restore intratumoral anti-tumor T cell responses and enhance the efficacy of anti-PD-1 by increasing active DCS in both mouse OSCC and CRC syngeneic tumour models. Direct interference with CAF-derived Wnt2 restored DC differentiation and DC-mediated antitumor T cell responses [14]. Combined with our experimental results, we consider whether targeting *Wnt2* can enhance the efficacy of ICI in *BRAF*-mutanted CRC. The specific results need to be further explored. The specific mechanism of the significantly high *Wnt2* expression in *BRAF*-mutated CRC is still unclear. One study exploring the treatment of *BRAF*-mutated CRC using a xenograft model, found that both pyrvinium and axitinib were able to significantly increase the ability of vemurafenib to attenuate tumor growth in xenografts of *BRAF*-mutated colorectal cancer cells. Thus, this also demonstrated that Wnt treatment also has non-immune-specific effects on BRAF-mutated CRCs [36]. 

This study is the first to report that *Wnt2* is significantly higher in *BRAF*-mutated CRC tissues than in wild-type tissues, which provides a new approach for exploring the development and treatment of patients with *BRAF* mutations.

Our study also has some limitations. Firstly, this study is limited by small sample size and single-center data; we did not separate the TNM stages for I, II, III and IV. In future studies, a multi-center large-scale study will be conducted. Secondly, the molecular mechanisms linking *Wnt2* and *BRAF* mutations require further exploration.

## 5. Conclusions

The results of this study suggest that high *Wnt2* expression and *BRAF* mutations could be used as new predictors of CRC prognosis. In addition, high *Wnt2* expression and *BRAF* mutations may be potential new therapeutic targets and have important implications in the future treatment of CRC. *Wnt2* expression is significantly higher in *BRAF*-mutated CRC than the wild-type, and *Wnt2* can be used as a new therapeutic target for *BRAF*-mutated CRC patients.

## Figures and Tables

**Figure 1 medicina-59-01133-f001:**
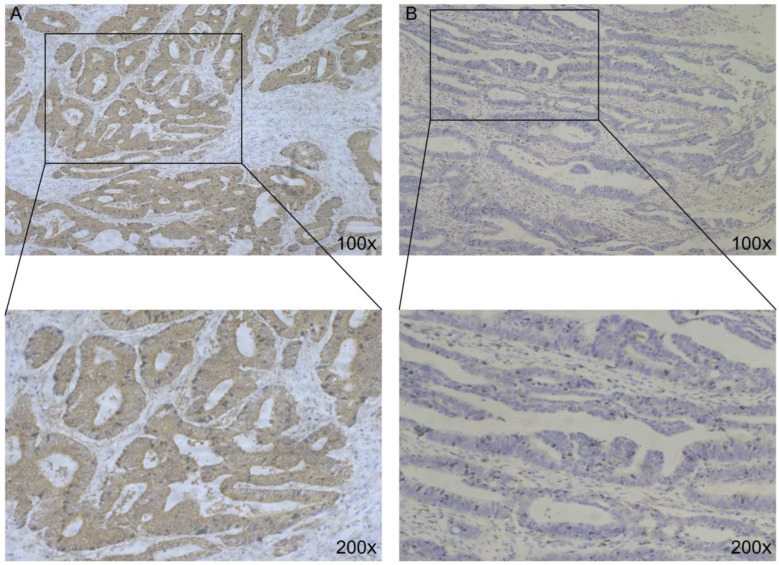
*Wnt2* expression in CRC. (**A**) High *Wnt2* expression; (**B**) low *Wnt2* expression.

**Figure 2 medicina-59-01133-f002:**
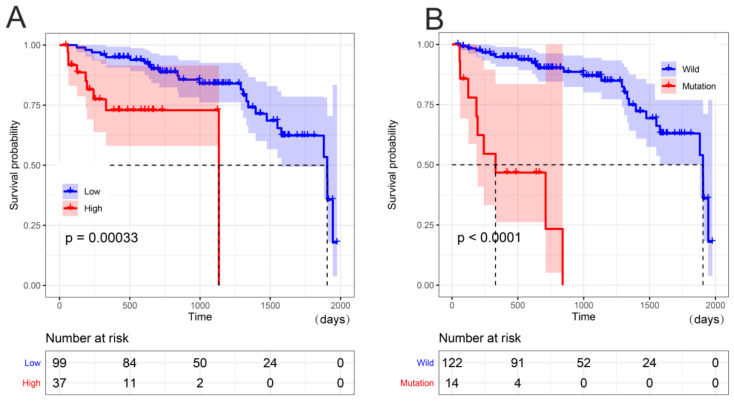
Survival analysis of CRC with high *Wnt2* expression and *BRAF* mutations. (**A**) Survival analysis of low *Wnt2* expression vs. high *Wnt2* expression in CRC; (**B**) survival analysis of *BRAF* mutations vs. wild type in CRC. *p* < 0.05 was considered statistically significant.

**Figure 3 medicina-59-01133-f003:**
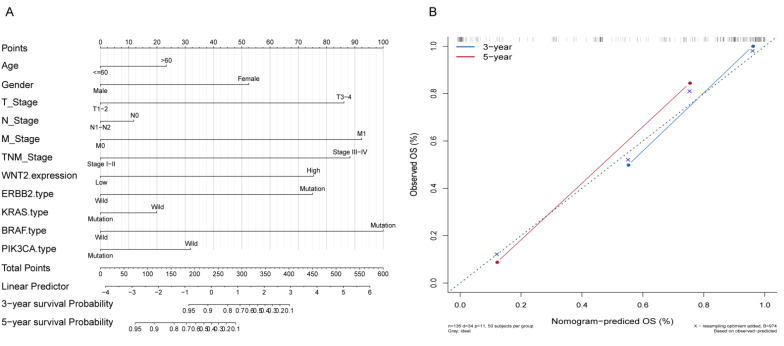
Prediction of OS after CRC resection. (**A**) Prognosis based on clinicopathological features and *Wnt2* expression; (**B**) time-dependent receiver operating characteristic curve of prognosis score based on combined *Wnt2* expression combined with clinicopathological variables. Perfect prediction would correspond to the dotted line. The pionts represent the worst and best survival of the patients.

**Figure 4 medicina-59-01133-f004:**
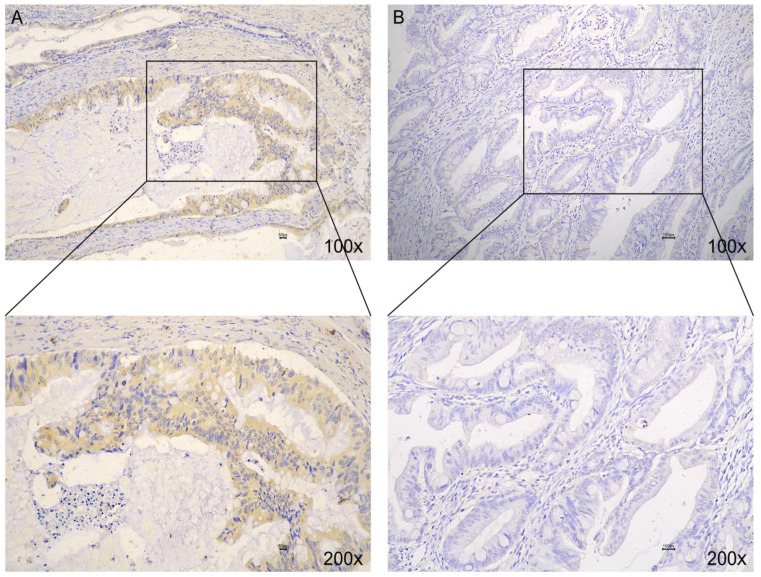
*Wnt2* expression in *BRAF*-mutated CRC. (**A**) High *Wnt2* expression; (**B**) low *Wnt2* expression.

**Table 1 medicina-59-01133-t001:** The mutation detection sites of the 136 samples.

Exon	Base Changes
*KRAS-2*	35G > A
	34G > A
	35G > C
	35G > T
	34G > C
	34G > T
	37G > T
*BRAF-15*	1799T > A
*PIK3CA-9*	545G > A
	542G > A
*PIK3CA-20*	1047A > G
*ERBB2-20*	2369C > T

**Table 2 medicina-59-01133-t002:** Correlation between *Wnt2* expression and clinicopathological features in CRC.

Clinicopathological Characteristics	Total	High Expression (*n* = 37)	Low Expression (*n* = 99)	*p*-Value
Age				0.58
≤60	34	11	23	
>60	102	26	76	
Gender				0.89
Female	63	18	45	
Male	73	19	54	
T-Stage				0.65
T1-2	35	8	27	
T3-4	101	29	72	
N-Stage				0.63
N0	87	22	65	
N1-N2	49	15	34	
M-Stage				0.69
M0	113	32	81	
M1	23	5	18	
TNM-Stage				0.99
Stage I-II	82	22	60	
Stage III-IV	54	15	39	
*ERBB2*				
Wild-type	132	36	96	1
Mutation	4	1	3	
*KRAS*				
Wild-type	83	23	60	1
Mutation	53	14	39	
*BRAF*				
Wild-type	122	26	96	0.0001
Mutation	14	11	3	
*PIK3CA*				
Wild-type	112	30	82	0.8
Mutation	24	7	17	

**Table 3 medicina-59-01133-t003:** Univariate and multivariate analysis of prognostic factors in CRC.

Variable	Univariate Analysis		Multivariate Analysis	
	HR (95%CI)	*p*-Value	HR(95% CI)	*p*-Value
Age	0.48 (0.22–1.07)	0.074		
Gender	0.71 (0.35–1.44)	0.338		
*Wnt2* expression	6.3 (2.43–16.33)	<0.0001	4.19 (1.1–15.95)	0.035
T-Stage	15 (2.02–111.36)	0.008	6.51 (0.77–54.62)	0.085
N-Stage	7.16 (3.03–16.93)	<0.0001	0.58 (0.11–3.01)	0.519
M-Stage	6.27 (3.01–13.07)	<0.0001	2.56 (0.99–6.65)	0.053
TNM-Stage (III–IV v.s. I–II)	9.91 (3.73–26.3)	<0.0001	4.8 (0.61–37.73)	0.136
*ERBB2*	0.39 (0.05–2.98)	0.366		
*KRAS*	0.88 (0.43–1.81)	0.724		
*BRAF*	15.96 (6.17–41.3)	<0.0001	7.12 (1.94–26.14)	<0.0001
*PIK3CA*	0.87 (0.35–2.16)	0.77		

**Table 4 medicina-59-01133-t004:** Correlation between *BRAF* mutations and *Wnt2* expression assessed by immunochemistry in CRC.

*BRAF* Mutation	*Wnt2* Expression		Tatol, *n*	χ2	*p*
	High	Low			
Positive	41	9	50	52.672	<0.001
Negtive	26	92	118		
Total, *n*	67	101			

## Data Availability

The data presented in this study are available in this article.

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
