# Peer review of "High Wnt2 Expression Confers Poor Prognosis in Colorectal Cancer, and Represents a Novel Therapeutic Target in BRAF-Mutated Colorectal Cancer"

_medicina, 2023, doi:10.3390/medicina59061133_

Round 1

Reviewer 1 Report

The article "High Wnt2 expression confers poor prognosis in colorectal cancer, " represents a novel therapeutic target in BRAF-mutated colorectal cancer“ represents an interesting and valuable original scientific paper describing the impact of Wnt2 expression in CRC patients with BRAF 1799T > A mutation.

The entire manuscript is well-written, adequately illustrated, and accompanied by an appropriate reference list.

However, some minor obstacles must be resolved before the manuscript is published.

These, among others, include:

The catalog numbers of the primary antibody used in the study should be mentioned in the manuscript text.

The names of the physicians from the Department of Pathology involved in WNT2 expression scoring, if stated as coauthors,  should be mentioned in the manuscript text.

The authors have written: “The samples were 139 FFPE specimens from postoperative CRC patients, and DNA was isolated and extracted, followed by PCR analysis”. However, the number of patients included in the study was 136 (“In total, 136 patients diagnosed with CRC who underwent surgical resection at the Affiliated Hospital of Jiangnan University between June 2014 and November 2016 were selected retrospectively.”). This should be explained, or the numbers correct.

The procedure used for DNA extraction and PCR reaction should be explained.

The table of five mutant genes should be enumerated and mentioned in the text.

In the subsection Clinicopathological features of the patients with CRC, the authors should mention the table with the corresponding data. 

In Table 1, the authors should choose between the terms wild and wild type and use one, not both.

The authors have written: “The difference in Wnt2 protein expression (high vs. low) was significantly correlated with BRAF mutation status in patients with CRC (P < 0.05), as shown in Table 1.”. The exact P value  (0.0001) should be stated.

The authors have written: “Subsequently, all important variables were entered into the multivariate Cox proportional hazards model, and the results showed that Wnt2 expression (P = 0.035) and BRAF mutations (P < 0.0001) were prognostic factors for poor OS in CRC (Table 2).” The term all important variables should be replaced with their names or the term variables statistically significant in univariate analysis.

A minor revision is suggested.

Author Response

  1. The catalog numbers of the primary antibody used in the study should be mentioned in the manuscript text.

Response: Thank you very much for your comments and suggestions. We have added the catalog numbers of the primary antibody to the manuscript.

  1. The names of the physicians from the Department of Pathology involved in WNT2 expression scoring, if stated as coauthors,  should be mentioned in the manuscript text.

Response: Thanks for your advice, we have thanked the pathology subject physicians involved in the Wnt 2 expression scoring in the acknowledgements at the end of the manuscript. 

  1. The authors have written: “The samples were 139 FFPE specimens from postoperative CRC patients, and DNA was isolated and extracted, followed by PCR analysis”. However, the number of patients included in the study was 136 (“In total, 136 patients diagnosed with CRC who underwent surgical resection at the Affiliated Hospital of Jiangnan University between June 2014 and November 2016 were selected retrospectively.”). This should be explained, or the numbers correct.

Response: Thank you very much for your review and discovery, this is a mistake and carelessness in our manuscript. We have corrected and seriously checked the number. Thank you again.

  1. The procedure used for DNA extraction and PCR reaction should be explained.

Response: Thank you very much for your comments, this is our negligence, and we have added a detailed description of the procedure of DNA extraction and PCR reaction:

DNA extraction of the samples was performed using a paraffin tissue DNA extraction kit(TianGen,beijing,China), the procedure is as follows: 30mg of paraffin tissue was collected, deparaffinized by shaking with xylene, and centrifuged at 12000g for 2min at room temperature. The supernatant was discarded, and absolute ethanol was added and mixed by shaking. The mixture was centrifuged at 12000g for 2min at room temperature, and the supernatant was discarded. It was allowed to stand for 5 minutes to fully volatilize the ethanol. An additional 200ul GA buffer and 20ul Proteinase K were added, mixed, and incubated at 56 ° C for 1h until the sample was completely lysed. After further incubation at 90℃ for 1h, 220ul GB buffer was added and mixed, then 250ul absolute ethanol was added and mixed. The liquid was put into the adsorption column at 8000rmp and centrifuged at room temperature for 2min. After the waste liquid was discarded, 500ul GD buffer was added to the adsorption column CR2 and centrifuged at 8000rpm for 60sec. Discard the waste solution and repeat twice. The adsorption column was opened for 5min, and then 50ul of TE eluate prewarmed at 65 ° C was added. Finally, DNA was collected.

Tissue samples were tested for mutations by Q-PCR , the procedure is as follows: the DNA of the extracted tissue samples was mixed with LMG mixed enzyme in the kit, and then the machine was detected. The reaction program is as follows: the first stage :42℃ for 5min, 95℃ for 5min, a cycle; Second stage :95 ° C 25s, 64 ° C 20s, 72 ° C 20s, 10 cycles: Third stage :93 ° C 25s, 60 ° C 35s, 72 ° C 20s, 36 cycles.

  1. The table of five mutant genes should be enumerated and mentioned in the text.

Response: Thanks for your reminder, we have added relevant content to the manuscript.

  1. In the subsection Clinicopathological features of the patients with CRC, the authors should mention the table with the corresponding data. 

Response: Thanks for your comments, we have added relevant content to the manuscript.

  1. In Table 1, the authors should choose between the terms wild and wild type and use one, not both.

Response: I would like to express my heartful thanks for your opinion. This is our mistake. I have changed the wild in the table to wild-type.

  1. The authors have written: “The difference in Wnt2 protein expression (high vs. low) was significantly correlated with BRAF mutation status in patients with CRC (P < 0.05), as shown in Table 1.”

Response: Thank you most sincerely for your comments, we immediately changed P <0.05 to P=0.0001 in the manuscript.

  1. The authors have written: “Subsequently, all important variables were entered into the multivariate Cox proportional hazards model, and the results showed that Wnt2 expression (P = 0.035) and BRAF mutations (P < 0.0001) were prognostic factors for poor OS in CRC (Table 2).” The term all important variables should be replaced with their names or the term variables statistically significant in univariate analysis.

Response: Thanks for your suggestions, we have added these important variables to the manusc.

Finally, I would like to express my gratitude to you again. Your detailed and serious comments and suggestions  have given us a great help. I hope our reply can get your approval.

Reviewer 2 Report

Overall, a very well-written paper. My only comment would be to perhaps provide some comments/speculations around why it is that a majority of the tumors with BRAF mutations have Wnt overexpression. Are there pathways shared by these two that link them together? Also, there are some publications out there that look at using WNT-targeted therapeutics in cell lines with BRAF mutations that should be commented on in the discussion. This paper in particular used xenograft models, so also demonstrate that WNT therapies also have a non-immune specific effect on BRAF mutated CRCs (i.e. Khanh B. TranSharada KolekarQian WangJen-Hsing ShihChristina M. BuchananSanjeev DevaPeter R. Shepherd; Response to BRAF-targeted Therapy Is Enhanced by Cotargeting VEGFRs or WNT/β-Catenin Signaling in BRAF-mutant Colorectal Cancer Models. Mol Cancer Ther 1 December 2022; 21 (12): 1777–1787. https://doi.org/10.1158/1535-7163.MCT-21-0941).

Author Response

Overall, a very well-written paper. My only comment would be to perhaps provide some comments/speculations around why it is that a majority of the tumors with BRAF mutations have Wnt overexpression. Are there pathways shared by these two that link them together? Also, there are some publications out there that look at using WNT-targeted therapeutics in cell lines with BRAF mutations that should be commented on in the discussion. This paper in particular used xenograft models, so also demonstrate that WNT therapies also have a non-immune specific effect on BRAF mutated CRCs (i.e. Khanh B. Tran, Sharada Kolekar, Qian Wang, Jen-Hsing Shih, Christina M. Buchanan, Sanjeev Deva, Peter R. Shepherd; Response to BRAF-targeted Therapy Is Enhanced by Cotargeting VEGFRs or WNT/β-Catenin Signaling in BRAF-mutant Colorectal Cancer Models. Mol Cancer Ther 1 December 2022; 21 (12): 1777–1787.

Response:Sincerely thank you for your comments, very sorry that we ignored this important paper. Thanks for your recommendation, we read this article carefully, quoted it in the discussion, and added some meaningful discussion based on the content.

Details are as follows: The specific mechanism of the significantly high Wnt2 expression in BRAF-mutated CRC is still unclear. One study exploring the treatment of BRAF-mutated CRC using a xenograft model, found that both pyrvinium and axitinib were able to significantly in-crease the ability of vemurafenib to attenuate tumor growth in xenografts of BRAF-mutated colorectal cancer cells. Thus also demonstrated that Wnt treatment also has non-immune specific effects on BRAF mutated CRCs[36]. 

Finally, I would like to express my gratitude to you again. Your detailed and serious comments and suggestions  have given us a great help. I hope our reply can get your approval.

I would like to express my gratitude to you.

Reviewer 3 Report

In this manuscript, the authors analyzed the protein expression of Wnt2 in some colorectal cancer by IHC and investigated their correlations with BRAF mutation. In conclusion, they find that High Wnt2 expression confers poor prognosis in colorectal cancer and was significantly associated with BRAF-mutated CRC. However, the current manuscript have some minor defects, and is also not carefully prepared. All these should be revised before the manuscript become to be publishable. The following are some comments for the manuscript in detail. 

1.     In the IHC staining results, about 2/3 of specimen showed low expression levels of Wnt2, indicated that Wnt2 expression is always low in CRC. This is conflicted with the oncogenic role of Wnt2, which should be highly expressed in tumors. The authors should explain why they get this result.

2.     The authors cited many references to show that both Wnt2 and BRAF promotes metastasis. However, the did not find any associations of Wnt2 or BRAF mutation with the clinicopathological features related to the metastasis, such as N-stage, M-stage. The also did not analyze the distal metastasis free survival.  

3.     It is not clearly state whether the 50 samples of BRAF-mutated CRC included the 14 samples of BRAF-mutated CRC in the first 136 samples.  

4.     In the 2nd and 3rd papragraph of introduction, the font of some sentences was changed. It should be corrected.

Author Response

  1. In the IHC staining results, about 2/3 of specimen showed low expression levels of Wnt2, indicated that Wnt2 expression is always low in CRC. This is conflicted with the oncogenic role of Wnt2, which should be highly expressed in tumors. The authors should explain why they get this result.

Response: First of all, thank you very much for raising such a meaningful question. According to our results retrieved from the TCGA-COAD and READ, Wnt 2 was indeed highly expressed in CRC, which was considered as a pro-oncogogenic gene. But Wnt 2 expression showed a dispersed trend in CRC. Wnt 2 proteins have certain expression in CRC, but our criteria of high and low grouping are different. Later, if  Wnt 2 is used as a diagnostic marker, it may be divided by strong positive and weak positive to replace the description of high and low expression. This study mainly observed the relationship between Wnt 2 expression and BRAF mutated CRC, and explored whether Wnt 2 could be a new therapeutic breakthrough point for BRAF-mutated CRC.

  1. The authors cited many references to show that both Wnt2 and BRAF promotes metastasis. However, the did not find any associations of Wnt2 or BRAF mutation with the clinicopathological features related to the metastasis, such as N-stage, M-stage. The also did not analyze the distal metastasis free survival.  

Response: This is a very meaningful opinion and thank you very much for your reminder. First, the correlation between wnt 2 and N-stage, M-stage was reflected in Table1.Then we analyzed the correlation between BRAF mutation status and N-stage, M-stage, and found P> 0.05 with no obvious significance, and the results are shown in the table below. Secondly, because the relevant data of DMFS is still in follow-up, it is not available. We are very sorry for this, and we will continue to track the relevant data in the future.

Clinicopathological characteristics

Total

BRAF

Wild-type

(n = 14)

BRAF

Mutation

(n = 122)

P-value

N-Stage

0.057

N0

87

5

82

N1-N2

49

9

40

M-Stage

0.69

M0

113

10

103

M1

23

4

19

  1. It is not clearly state whether the 50 samples of BRAF-mutated CRC included the 14 samples of BRAF-mutated CRC in the first 136 samples.  

Response: Thanks for your comments, which is our oversight, the 50 samples of BRAF-mutated CRC contained the 14 samples of BRAF-mutated CRC from the previous 136 samples, which we annotated in the manuscript.

  1. In the 2nd and 3rd papragraph of introduction, the font of some sentences was changed. It should be correct

Response: Thanks again for your valuable comments and we apologize for our carelessness and correct that.

Thank you most sincerely for your comments.
